# Using the app "Injurymap" to provide exercise rehabilitation for people with acute lateral ankle sprains seen at the Hospital Emergency Department–A mixed-method pilot study

Jonas Bak[1], Kristian Thorborg[2,3,4], Mikkel Bek Clausen[2,5], Finn Elkjær Johannsen[6,7], Jeanette Wassar Kirk[1,8], Thomas Bandholm[1,2,3,4]*

1 Department of Clinical Research, Copenhagen University Hospital, Amager and Hvidovre, Copenhagen, Denmark, 2 Department of Orthopedic Surgery, Copenhagen University Hospital, Amager and Hvidovre, Copenhagen, Denmark, 3 Department of Clinical Medicine, University of Copenhagen, Copenhagen, Denmark, 4 Physical Medicine & Rehabilitation Research–Copenhagen (PMR-C), Department of Physical and Occupational Therapy, Copenhagen University Hospital, Amager and Hvidovre, Denmark, 5 Department of Midwifery, Physiotherapy, Occupational Therapy and Psychomotor Therapy, Faculty of Health, University College Copenhagen, Copenhagen N, Denmark, 6 Institute of Sports Medicine Copenhagen, Copenhagen University Hospital, Bispebjerg and Frederiksberg, Copenhagen, Denmark, 7 Injurymap Aps, Copenhagen N, Denmark, 8 Department of Health and Social Context, National Institute of Public Health, University of Southern Denmark, Odense, Denmark

* thomas.quaade.bandholm@regionh.dk

**Data Availability Statement:** Public deposition of raw data points would breach compliance with the

## Abstract

### Background

Acute lateral ankle sprains (LAS) account for 4–5% of all Emergency Department (ED) visits. Few patients receive the recommended care of exercise rehabilitation. A simple solution is an exercise app for mobile devices, which can deliver tailored and real-time adaptive exercise programs.

### Purpose

The purpose of this pilot study was to investigate the use and preliminary effect of an app-based exercise program in patients with LAS seen in the Emergency Department at a public hospital.

### Materials and methods

We used an app that delivers evidence-based exercise rehabilitation for LAS using algorithm-controlled progression. Participants were recruited from the ED and followed for four months. Data on app-use and preliminary effect were collected continuously through the exercise app and weekly text-messages. Baseline and follow-up data were collected though an online questionnaire. Semi-structured interviews were performed after participants stopped using the app. Results: Health care professionals provided 485 patients with study information and exercise equipment. Of those, 60 participants chose to enroll in the study and 43 became active users. The active users completed a median of 7 exercise sessions.

protocol approved by the Danish Data Protection Agency. In the paper, we have also had to limit the number of indirect identifiers in cases where there might be a risk of compromising patient privacy (e.g. Tables 1 and 2). Reviewers and others may obtain access to the data by request, and after a data processing agreement has been signed. There will be no limitation to data sharing as long as a data sharing agreement is signed. As per Danish research code, we are required to store research data for a minimum of 5 years starting from the time of publication. We will ensure this by standard data management and storage at our institution. Please address any correspondence to Dr Thomas Bandholm; thomas.quaade.bandholm@regionh.dk (corresponding author) or the Department of Clinical Research, Hvidovre Hospital, Kettegaard Alle 30, DK-2650 Hvidovre, Copenhagen, Denmark. Phone: +45 3862 3862.

**Funding:** The study was funded partly by the European Regional Development Fund (https://ec.europa.eu/regional_policy/funding/erdf_en), which was administered by the Copenhagen Center for Health Technologies (CACHET, https://www.cachet.dk/) and given to TB. Funding was for salary support for a research assistant (JB). The funders had no role in study design, data collection and analysis, decision to publish, or preparation of the manuscript (please see details in the conflict-of-interest statement).

**Competing interests:** I have read the journal's policy and the authors of this manuscript have the following competing interests: FEJ is a co-founder of Injurymap. This conflict was accommodated by restricting FEJ from any deciding role in terms of study design, study management, data interpretation, report writing and submission. TB has received speaker's honoraria for talks or expert testimony on the efficacy of exercise therapy to enhance recovery after surgery at meetings or symposia held by biomedical companies (Zimmer Biomet and Novartis). He is an editorial board member with Br J Sports Med. KT is a deputy editor with Br J Sports Med and receives an annual honoraria. JWK, JB and MBC have declared that no competing interests exist.

Most of the active users were very satisfied or satisfied (79%-93%) with the app and 95.7% would recommend it to others. The interviews showed that ankle sprains were considered an innocuous injury that would recover by itself. Several app users expressed they felt insufficiently informed from the ED health care professionals. Only 39% felt recovered when they stopped exercising, and 33% experienced a recurrent sprain in the study period. Conclusion: In this study, only few patients with LAS became active app users after receiving information in the ED about a free app-based rehabilitation program. We speculate the reason for this could be the perception that LAS is an innocuous injury. Most of the patients starting training were satisfied with the app, although few completed enough exercise sessions to realistically impact clinical recovery. Interestingly more than half of the participants did not feel fully recovered when they stopped exercising and one third experienced a recurrent sprain.

## Trial-identifiers

https://clinicaltrials.gov/ct2/show/NCT03550274, preprint (open access): https://www.medrxiv.org/content/10.1101/2022.01.31.22269313v1.

## Introduction

An ankle sprain is one of the most common musculoskeletal injuries with a comprehensive burden for individuals as well as society [1]. They account for 4–5% of all Emergency Department visits in Denmark [2] which is consistent with data from other countries [1]. This might just be the tip of the iceberg since less than half of the people who sustain an ankle sprain seek health care [1,3]. A lateral ankle sprain (LAS) is often regarded as an innocuous injury [4] and especially health care professionals tend to overestimate the recovery [5]. However, 32–74% of people who sustain a LAS have prolonged symptoms such as pain, decreased function and subjective instability for several years after their initial injury [1,6]. In sport, up to 34% will sustain a recurrent sprain in the following years after their initial injury [6]. However, exercise therapy is a well-documented cost-effective rehabilitation modality to treat LAS [7–10] and prevent recurrent sprains [8,11,12]. It is unfortunate that few patients are prescribed exercise programs or physiotherapy after a LAS, and that most expenses relate to diagnostic procedures rather than exercise-based rehabilitation [4]. Technologically supported self-management may be a solution to this problem but requires investigation.

Applications for smart devices (apps) have the potential to be powerful tools in providing easily accessible exercise therapy programs and in attaining important information about exercise behavior that have been practically unobtainable previously [13]. They represent a flexible telehealth solution that most often does not require the online presence of a health care professional. Apps have the ability through interaction between users and smartphone to tailor specific information and exercise programs. Furthermore, apps can give real-time, real-life feedback during an exercise session without waiting for a health specialist to be available [13]. A serious challenge in the use of apps, however, is that there is a general lack of evidence-based solutions, and that health apps often wrongly claim to be evidence founded [14–16]. Another challenge is that the effectiveness of the majority of health apps are fairly unknown and their ability to make actual behavioral change is often poorly reported [17,18]. The high availability of health apps on a unregulated market poses a major concern since it may have a major influence on rehabilitation success or at worst cause harm [14].

Injurymap is an exercise app designed for treating different musculoskeletal problems including LAS. The Injurymap exercise program has been developed by health care professionals and has the potential to provide an easy-accessible management of rehabilitation. However, the app has currently not been tested in a clinical study. Before undertaking a large-scale study, we wish to pilot test the app to assess the use and preliminary effect of the app-based exercise program. We are particularly interested in the proportion of patients who become active app users (and for how long) when provided with an option of a free app-based rehabilitation program in an acute care clinical setting.

### Purpose statement

The purpose of this pilot study was to investigate the use and preliminary effect of an app-based exercise program in patients with LAS seen in the Emergency Department at a public hospital. Consistent with the mixed-method study design outlined below, both quantitative and qualitative outcomes were collected to fully explore factors related to uptake and adherence to the exercise program.

## Method

### Study design

The study used an explanatory sequential mixed method cohort study design (QUAN→qual) [19]. The study was conducted in subsequent phases, with the development of the quantitative outcomes leading the following development of qualitative outcomes, and the quantitative app-use data guiding a purposive sampling for semi-structured interviews [20]. Quantitative and qualitative outcomes were analyzed separately and integration was done in the interpretation process by triangulation (sections with integration: Methods, Results, and Discussion) [21]. The study process is outlined in Fig 1. We consider the study exploratory and, hence, it was designed with a flat outcome structure using multiple evenly valued outcome measures.

Outcomes related to "app-use" were quantitative data collected directly from the app during the exercise period and after by a follow-up questionnaire. Outcomes meant to provide explanatory insight of the app-use data were qualitative and collected by semi-structured interviews after the exercise period. Outcomes related to "preliminary effect" were quantitative clinical recovery-data collected using weekly TEXT-MESSAGES for four months after the initial injury.

The SPIRIT checklist [22] and PREPARE Trial guide [23] were used to develop the study protocol. The Danish National Committee on Health Ethics approved this study (ID: 17041467). All personal data were handled according to the Danish act concerning processing of personal data. Prior to the study, Injurymap had been approved for handling personal data by the Danish Data Protection Agency (file no. 2016-42-3535). The study followed the principles of the Helsinki declaration and it was pre-registered at ClinicalTrials.gov (https://clinicaltrials.gov/ct2/show/NCT03550274). We report the study using the Good Reporting of A Mixed Methods Study (GRAMMS) checklist [24] (S1 Table).

### Study setting

The study was performed at a Danish public hospital where patients are covered by the Danish healthcare system. At the hospital, the current practice for non-surgical management of LAS patients is RICE (Rest, Ice, Compression, Elevation), mobility exercises and recommendations of slowly returning to activity. The current practice does not involve any on-site systematic instruction in evidence-based rehabilitation programs or referral to such elsewhere. Therefore,

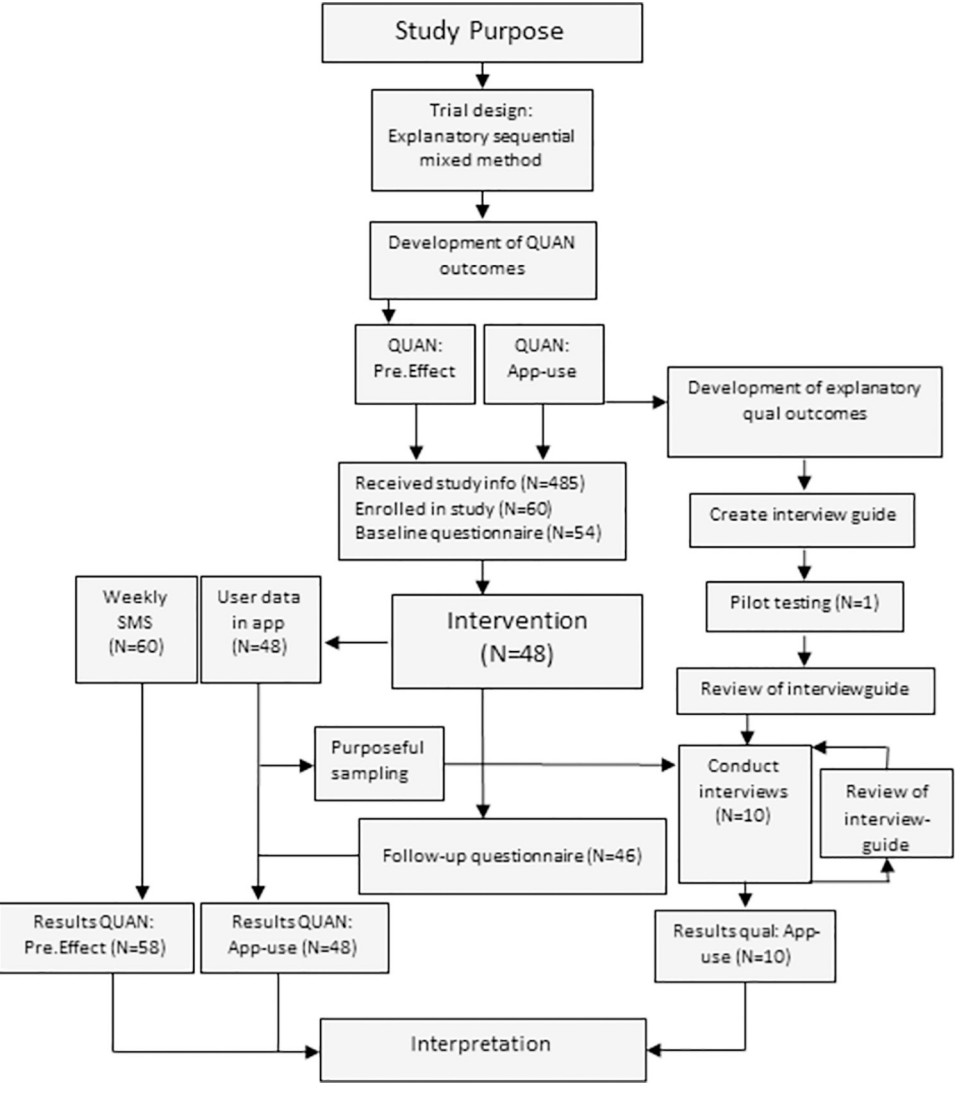

**Fig 1. Mixed-method study process.**

we wanted to explore the use of a simple app-based solution in this setting. Informed written consent from the participants were registered before participants started the app-based exercise program. Participants could withdraw from the study at any time, without any consequences.

## Participants

Participants were recruited from the Emergency Department at Copenhagen University Hospital, Hvidovre. They were asked to participate in a home-based rehabilitation delivered by an app (Injurymap) available on any smart device. The inclusion criteria was; patients with an acute lateral ankle sprain (< 48 hours from injury) diagnosed by a relevant health care professional at the hospital ED. Gradings of ankle sprain severity was not performed, since this is not standard procedure in the ED. The exclusion criteria were; concurrent fracture of the leg or foot (Ottawa rules and/or x-ray), previous surgery in the ankle or surgery as a consequence of the current ankle sprain, serious illness (terminal patient, rheumatoid arthritis, fibromyalgia etc.), not owning a smart device (phone or tablet) or unable to understand and read Danish.

## Procedures

Health care professionals associated with the ED and responsible for ankle examinations recruited participants. When a health care professional identified a patient with ankle sprain, they delivered a recruitment bag containing several rubber bands of different thickness and a description of the free app-based exercise opportunity in this study. This approach was chosen to resemble a delivery method applicable in clinical practice. Besides the written information, the health care professionals were encouraged to recommend the exercise program to the patients. If a patient was willing to participate in the project, they contacted the research assistant (JB) by the contact information in the written material. When contact was established, and participants were deemed eligible, they received a voucher for free access to the app program, informed consent, and a baseline questionnaire.

The project was implemented at the ED by the primary investigator (JB). Health care professionals in the ED were informed about the study at staff meetings, by the weekly newsletter, and by the primary investigator who participated in the daily routines prior to recruitment for the project. Two large boxes containing the recruitment bags were placed strategically in the ED office and the primary examination room. The boxes had a large picture of an ankle sprain at the front and a text asking to give patients an exercise opportunity. This recruitment procedure was chosen to reflect a normal clinical care setting and to encompass both the health care professionals' willingness to promote the app solution to patients with LAS, as well as participants' willingness to accept the offer. For the same reason, we tried to inform about the study as being a rehabilitation exercise opportunity more than a research study.

## Intervention

InjuryMap offers exercise programs for LAS and other musculoskeletal conditions. The app requires user-registrations and a monthly paid subscription fee to access the exercise program. In this study, the app company provided free access for participants to the LAS exercise program. Examples from the app content can be seen in Fig 2.

The exercise program was available on any mobile device and/or tablet using Android or iOS operating systems. Participants could perform the exercises at any preferred location and were not restricted from seeking additional care. After four months of training or if a participant was inactive in the app for more than two consecutive weeks, they were considered to have stopped the exercise intervention.

The exercise program for LAS consisted of three phases with increasing difficulty. Each phase consisted of four categories of exercise. The categories were 1) mobility, 2) stability/balance, 3) strength and 4) stretching. The app could adjust the difficulty for each individual exercise depending on user-feedback. The user is asked to rate each exercise on a pain scale and difficulty scale after completing it. Their rating is used in an algorithm to calculate progression/regression for each exercise and to catch red flags. A comprehensive description of the exercises can be found in S1 Text and S2 Table.

The exercise program was set up so that several exercises were completed after each other to successfully complete a training session. Each exercise was accompanied by a video with an explanatory voice-over and the number of required repetitions written on the display. After completing an exercise, the participants registered pain level and difficulty of performing the exercise. If the participants registered no or low pain and low exercise difficulty, the app chose a progression of the exercise for the next exercise session. The app recommended participants to complete exercise sessions three times a week after app registration. If the participants followed the recommended three sessions a week, they would be able to reach the highest difficulty level in two to four months depending on their pain and difficulty answers. There were

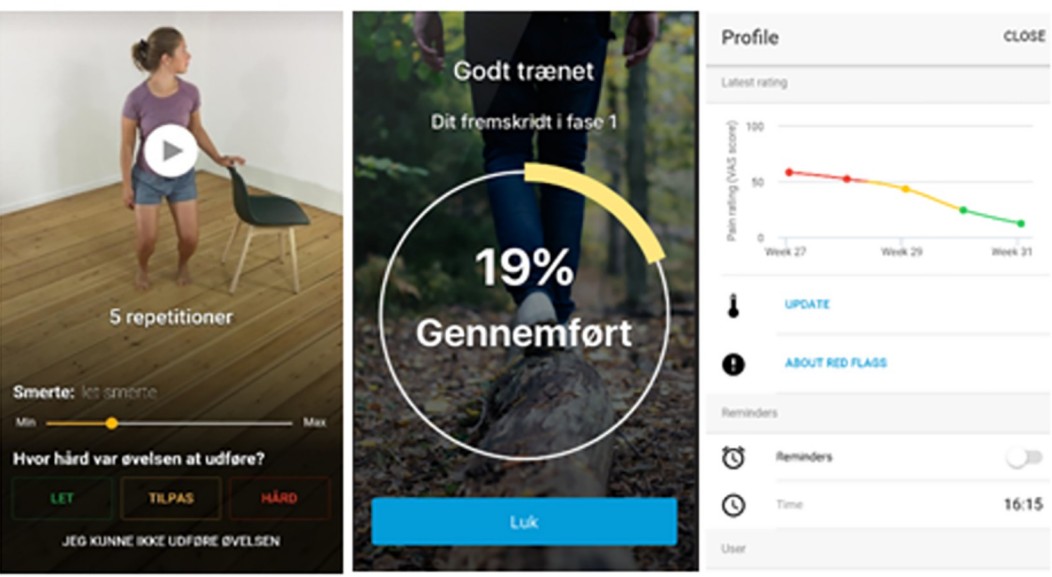

**Fig 2. Example of app content.**

no limitations in how many exercise sessions could be performed per week or a maximum number of weeks they could exercise except for the strengthening exercises which could only be performed once per day. Participants were able to activate a reminder function so that the app would remind them to exercise on a daily timepoint of their choosing. The spoken and written language in the application was Danish.

In the exercise program, circular rubber bands were used in several exercises. The rubber bands are common and cheap products available in most sports stores. In this study, the recruitment bag contained a selection of rubber bands with varying resistance.

## Outcomes

In this study, the outcome data were divided into two categories reflecting the study objective. The first category ("App-use") consists of quantitative data on uptake, retention, and adherence. Furthermore, it contained user-experience which is comprised of quantitative data on satisfaction with the app and qualitative data on the factors that influenced the app use. The second category ("Preliminary effect") consists of quantitative data on clinical recovery and recurrent injuries. A baseline questionnaire was completed after enrollment to gather descriptive data.

**App-use.** The overall rationale for the app-use outcomes below was to investigate how many participants exposed to the app that started using it; how much they used it; when they stopped using it; and how the user experience was. As a part of the app evaluation, we assessed uptake of the app-based exercise program. For app uptake, we calculated the following: number of participants diagnosed with an ankle sprain at the ED in the study period; number of participants who received a recruitment bag; number of participants willing to participate (contacted by the principal investigator); number of participants who became active users (defined as having downloaded the app and initiated the exercise program). By counting how many recruitment-bags the health care professionals delivered, the number of participants with ankle sprain who had been informed about the exercise opportunity could be estimated. From the Danish National Patient Register it was possible to obtain the number of ankle

sprains diagnosed at the ED. For retention, following were calculated: Number of participants completing baseline and follow-up questionnaire; and number of text-messages responded through the study period.

The app had mandatory user-registration so all exercise activity in the app was registered at the individual participant level. From these data, we calculated the following adherence outcomes: Number of exercise sessions completed per participant; and completed exercise sessions per week. If a participant did not commence the exercise program within two weeks, when active users was inactive for two consecutive weeks, or if they were active in the app for four months from their initial injury, they were considered finalized and received the follow-up questionnaire containing user-experience with the app-based exercise solution. Satisfaction on a five-point Likert scale were assessed for active users for the following items: the difficulty and the progression of the exercise program (Difficulty), the content of the exercise program (Content), the results from the exercise program (Results), and the usability of the app (User-friendly). Furthermore, all participants were asked if they would recommend the app to others (yes/no) and how much they would be willing to pay for the app (DKK).

After ending the exercise intervention, a group of participants were contacted for semi-structured interviews. The interviews focused on understanding and explaining motivational factors or barriers that may influence the use of the app "Injurymap". The study used a purposeful sampling for the interviews as recommended for explorative mixed method studies [20]. The sampling of participants was based on different number of completed exercise sessions, different age groups, both men and women. We did this to capture get a broad perspective on the motivational factors or barriers from both those with many completed exercise sessions and those who dropped out early. Based on the sampling criteria, a pragmatic number of ten participants were selected (both men and women at different age groups).

The interviews were performed by phone by the principal investigator (JB). Participants for the semi-structured interviews were contacted by mail and phone at the same time as the follow-up questionnaire. An interview guide was developed by the principal investigator (JB) with supervision from a senior researcher experienced in qualitative research (JWK). The guide was pilot tested on a person with experience in the exercise app, but otherwise not involved in the study. Recordings from the pilot testing were examined by two researchers (JWK and JB) to improve the interview technique and evaluate coherence of the questions in the interview guide. After the first interview, the recordings were examined again by the senior researcher (JWK) the recordings were compared to the purpose statement to ensure it was adequately covered in the interviews. Changes in the interview guide from the first interview consisted primarily of merging separate themes, based on how the participants associated and described their experiences. We also added questions about the experience at the ED and its impact on app-use since this factor was mentioned as an influential factor.

Interviews were recorded and transcribed verbatim. The data were analyzed using a thematic approach as described by Castleberry [25]. The data were coded, and recurring phrases or words were grouped into basic themes by the principal investigator (JB). Themes and codes were compared to the transcribed interview by two researchers (JB and JWK) to ensure that coding was performed with the same consistency and true to the original statements. This was performed in several processes until agreement was achieved. In this process, basic themes were clustered into global themes. The initial interpretation was performed by the principal investigator (JB) and reviewed by a researcher (JWK). Finally, the qualitative results were discussed with the whole research group.

**Preliminary effect.** The overall rationale for the preliminary effect-outcomes outlined below was to investigate if exercise adherence, was related to clinical recovery. Clinical recovery was evaluated by self-reported evaluation of symptoms using a weekly string of text-

messages for four months after their initial injury. The following Clinical Recovery items were collected by text-messages: Not able to fully participate in work/study because of the ankle sprain (days); Return to sport (RTS), defined as not able to fully participate in sport because of the ankle sprain for participants who registered as being "sports active" in the baseline questionnaire (weeks); Recurrent lateral ankle sprains in the same ankle (number); Subjective feeling of ankle stability (0–10 points). From the follow-up questionnaire, the clinical recovery item: Subjective feeling of recovery (yes/no) was also collected.

A recurrent sprain was defined as an inversion episode on the same ankle as assessed in the ED. Recurrent sprains were divided into two groups; 1) Recurrent sprain with time-loss, defined as being unable to continue current activity and/or unable to participate in work/ sports activities the next day because of the ankle; 2) Recurrent sprain with no time loss was defined as able to continue with current activities and able to participate in sports/work activities the next day.

As the exercise program in the app is built with similar component as other evidence-based exercise programs [8], we expected no harms from the intervention. Nonetheless, participants received a text-message in the weekly string of text-messages where they could register any discomforts or injuries related to performing the exercise program.

## Materiel and outcome assessment

The baseline and the follow-up questionnaires were collected without assessor involvement through RedCap (Research Electronic Data Capture)—a browser-based software developed by Vanderbilt University. Participants received emails containing a link to their personal online questionnaire. The research team had access to app use data through a log-in to the Injurymap online database. The text-messages were collected using SMS-track—an online system used to send and receive standardized text-messages. Each week the SMS-track system sent the first of six questions to the participants and waited for an answer before the next question was sent. If a participant reported an ankle sprain, she or he received a phone call from the principal investigator (JB) to clarify the type of sprain.

The principal investigator (JB) performed the outcome assessment, follow-up assessment, data extraction and data analysis. Participants active in the app were contacted as little as possible to minimize any potential influence on their exercise behavior.

## Statistical analysis

Baseline characteristics are summarized using suitable descriptive statistics. Normal distribution was assessed using the Shapiro-Wilk test and Q-Q plots. For the app-use outcomes, recruitment rates, retention rates are presented in suitable descriptive tables. Adherence is summarized in total sessions per participant and exercise sessions per week during the intervention period. Registered Harms were addressed individually. Quantitative user-experience are summarized using descriptive statistics.

We planned to determine the preliminary effect of the app by examining the relationship between the exercise dose (adherence) and clinical recovery outcomes using linear or logistic regression models, depending on type of outcome. The models would include the clinical recovery outcome as the dependent variable and exercise dose as independent variable. However, as the weekly text-messages that contained questions on clinical recovery were answered more frequently by participants who were very active in the app, we chose not to conduct the analyses of how exercise adherence related the clinical-outcomes as they would be biased. Instead, we report the clinical recovery data using descriptive statistics. Analyses were done using MS Excel, R, and MS Word.

## Sample size

Approaches to sample size justification for studies that investigate preliminary effectiveness of interventions, such as pilot and feasibility trials, vary. One rule of thumb-approach is 12 per group for a pilot RCT [26]. We used a pragmatic sample size 60 participants for this study. It was based on the rationale that 30 out of 60 would download the app, start using the app, and start using the exercise program, based on previous experience with the app on low back pain patients. We figured a sample size of 30 app users would equate to two groups of 12 participants each, if pooled [26], plus 6 to account for some attrition.

## Results

60 participants were recruited during the period from July 3, 2018, to April 3, 2019. 43 of these stated that they participated in weekly sports activities (Sport active group). Accounting for half of the sprains, sports were the most frequently reported cause of injury. One third (n = 20) reported a previous ankle sprain in the same ankle. The sprains were equally divided between left and right site sprains. Baseline characteristics are presented in Table 1.

**Table 1. Baseline characteristics of participants with acute lateral ankle sprains (N = 60).**

| Item | Mean(SD) | N |
|---|---|---|
| **Age (yr)** | In their 30s | 60* |
| **Weight (kg)** | 76.60(19.85) | 56 |
| **Height (cm)** | 174.04(10.75) | 56 |
| | **n(%)** | **N** |
| **Women** | 36(64.3%) | 56 |
| **Injury site (Right)** | 29(51.8%) | 56 |
| **Previous (same) ankle sprain (yes)** | 20(35.7%) | 56 |
| **Sports active, (yes)** | 43(78.2%) | 55 |
| **Education level:** | | 56 |
| *Primary education* | 11(19.6%) | |
| *Upper secondary education* | 10(17.9%) | |
| *Vocational Education* | 4(7.1%) | |
| *Short higher education* | 1(1.8%) | |
| *Bachelors program* | 23(41.1%) | |
| *Master program or higher* | 6(10.7%) | |
| *Other* | 1(1.8%) | |
| **Physical demands on work:** | | 55 |
| *Mostly sitting* | 22(40%) | |
| *Equal sitting and walking* | 19(34.5%) | |
| *Mostly Walking* | 14(25.5%) | |
| **Activity when injured:** | | 56 |
| ***Work*** | 8(14.3%) | |
| ***Sports*** | 26(46.4%) | |
| ***Leisure*** | 22(39.3%) | |
| ***Other*** | 0 (0%) | |
| One or more element of the RICE-principle (yes) | 52(92.9%) | 56 |

*Data on age were collected via the written informed consent. All other data were collected via the baseline questionnaire. Wording chosen to limit the number of indirect identifiers.

**Table 2. Characteristics of the interviewed participants with acute lateral ankle sprains (N = 10).**

|                       | 1     | 2     | 3     | 3     | 5     | 6     | 7     | 8     | 9     | 10    |
|-----------------------|-------|-------|-------|-------|-------|-------|-------|-------|-------|-------|
| M/W                   | W     | W     | M     | M     | M     | W     | M     | M     | M     | W     |
| Age (yr)              | *     | *     | *     | *     | *     | *     | *     | *     | *     | *     |
| Weight (Kg)           | *     | *     | *     | *     | *     | *     | *     | *     | *     | *     |
| Height (cm)           | *     | *     | *     | *     | *     | *     | *     | *     | *     | *     |
| Education[a]          | PostG | PostG | Skill | Grad  | Grad  | Grad  | Grad  | Grad  | <High | <High |
| Employment            | Out   | Job   | Job   | Job   | job   | Job   | Out   | Job   | job   | Stud< |
| No sessions completed | 26    | 21    | 4     | 16    | 8     | 3     | 5     | 1     | 7     | 3     |

[a]Education: < High School (<high), High School (High), Skilled (Skill), Graduate (Grad), Post Graduate (PostG)
[b]Employment: Employees or self-employed (job), Unemployed (No), Not in the labor force (out), Student in high school or lower education level (Stud<), graduate student or higher education level (Stud>).
\* Removed to limit the number of indirect identifiers.

Characteristics of the 10 participants interviewed can be seen in Table 2. They had completed between 1 and 26 exercise sessions. 19 participants were contacted to achieve 10 interviews. One participant refused to participate in the interview due to lack of time. The remaining eight participants did not respond to the request.

From the coding process [25], three themes were deducted (see Table 3). Each theme was divided by several sub-themes. Theme I-II are characterized by a substantial amount of data and a clear link to the purpose of the interviews. Theme III is also characterized by a substantial amount of data and includes factors that were not directly linked to the app experience but with a potential impact on the use of the app. Quantitative adherence data and qualitative data are presented through joint display table as recommended by Guetterman et al. [27].

## App-use

**Uptake.** According to the Danish National Patient Register [28], a total of 1110 people were diagnosed with an ankle sprain in the ED during the study period (see Table 4). It should be noted that this does not only include isolated sprains or lateral sprains, which is why the number of people is considered an absolute maximum of potentially eligible participants. 485 people received the recruitment bags containing rubber bands and study information. Of

**Table 3. Themes and subthemes.**

| Themes                                    | Subtheme                                     |
|-------------------------------------------|----------------------------------------------|
| **I: Motivational factors**               | *Usability*                                  |
|                                           | *The app's exercise level and ability to adapt* |
|                                           | *Ankle symptoms and expectations*            |
|                                           | *Influence of work and leisure time*         |
| **II: Technology assisted exercise behavior** | *Process statistics*                     |
|                                           | *Reminder function*                          |
|                                           | *Exercise Comprehension*                     |
|                                           | *Views on the visual expression*             |
| **III: Factors of importance for start-up** | *Diagnostic and prognostic expectations*   |
|                                           | *Treatment and preventive expectations*      |
|                                           | *Provider integrity*                         |
|                                           | *The user as independent searcher*           |

**Table 4. Uptake of the app intervention.**

| Project stage | No of people | Percentage of total diagnosed | Percentage of those who received info |
|---|---|---|---|
| Diagnosed with ankle sprain in the ED | 1110 | 100% | - |
| Received study information | 485 | 44.7% | 100% |
| Enrolled in the study | 60 | 5.4% | 12.4% |
| Active users | 48 | 4.3% | 9.9% |

these, 60 contacted the principal investigator and were included. 48 became active users of the app, according to our definition.

**Retention.** Of the 60 participants enrolled, 54 (92%) answered the baseline questionnaire (one participant had a partial completion) and 46 (77%) answered the follow-up questionnaire. Two participants dropped out, one because he did not become an active app user and, hence, did not want to answer the SMS-string, one due to pregnancy.

For the 60 participants in the 17 weeks follow-up period, a total of 6120 SMSs could potentially have been answered. A total of 4387 answers were received (72%), with the highest frequency in week 1 (85.0%) and the lowest in week 13 (61.9%) (see Fig 3).

**Exercise adherence.** 48 participants became active users because they completed a minimum of 1 exercise session (see Fig 4). The median number of completed exercise sessions in the four months period was 5.5 and ranged from 0 to 68 completed sessions with the majority of the participants completing few or very few sessions (see S1 Fig for completed sessions at the individual level). An exploratory analysis of the adherence by education level, age, work, and sports active can be found in S2 Fig.

The interviews sought to gain an explanatory insight regarding factors that may have influenced uptake and adherence to the exercise program. Theme I "Motivational factors" describes factors directly linked to adherence by the participants. Theme II "Technology assisted exercise behavior" describes how participants viewed technological features that may have influenced adherence but was not directly connected by the participants. Theme III "Factors of importance for start-up" describes factors that may have influenced participants to become active users. See Table 3 for subthemes related to the themes and Table 5 for a joint display of the quantitative and qualitative findings.

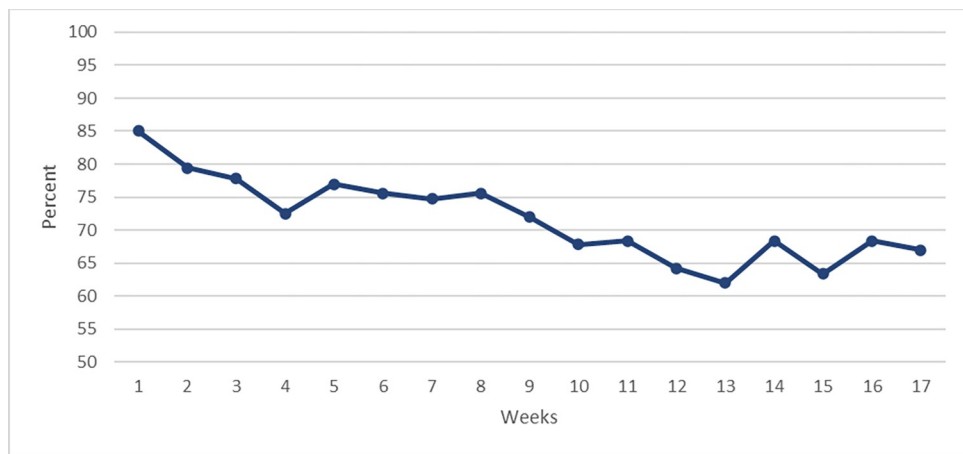

**Fig 3. Weekly SMS response rate.** Calculated as a mean percentage of the response percentages to the 6 SMS questions per week.

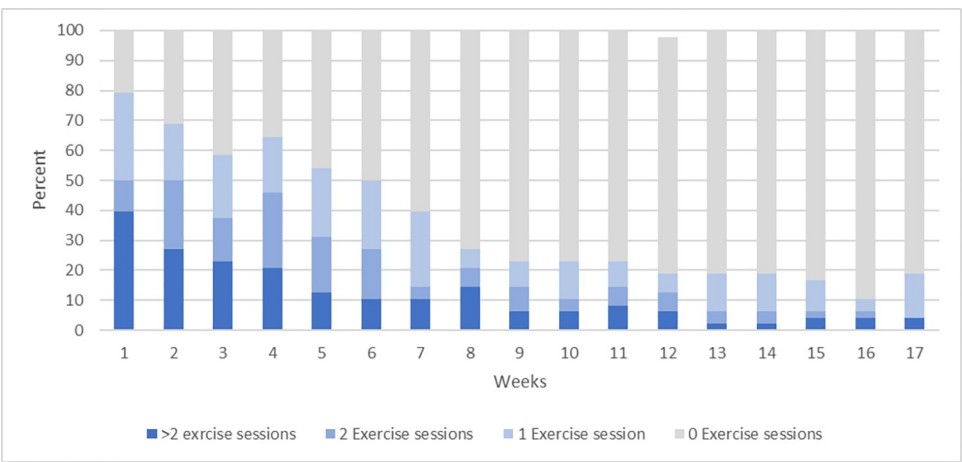

**Fig 4. Weekly distribution of completed exercise sessions.**

**Satisfaction.** When asked at follow-up, 95.7% of the participants would recommend the app to other people with an ankle sprain. 71% were willing to pay for the exercise program with an average cost of 46 DKK equivalent to 6.16 EUR. Satisfaction scores can be seen in Fig 5.

### Preliminary effect

A total of 36 recurrent sprains were reported in the follow-up period. Of these, 32 were time-loss injuries and 4 were not. 20 participants had minimum 1 recurrent sprain and 11 of the 20 had 2 or 3 recurrent sprains. No participants had more than 3 recurrent sprains in the period. At follow-up, 39.1% (18 participants) felt that the ankle was able to perform at the pre-injury level. The absent from work/study ranged from 0 to 100 days, the median being 1 (IQR = 6) day. In week 1 the average ankle stability was 4.5 (SD = 2.5) on the 0–10 scale and increased to 8.7 (SD = 1.5) in week 17. Weekly changes can be seen in Table 6. The sports active group (n = 43) had in average 9 weeks (SD = 4.9) where they could not participate in sports activities without restrictions from their sprained ankle. After 17 weeks, 30.2% still reported that they were restricted by their ankle in sports activities.

### Harms

No harms were registered.

### Discussion

The present study investigated the use of an app-based, rehabilitation exercise program for lateral ankle sprains by collecting data on use of the app "Injurymap".

From the 1110 patients who were diagnosed with an ankle sprain at the ED during the study period, 45% received the information about the free app-based exercise program. Of those who received the information, 10% became active users. In this study, we were able to determine exactly how many received study information and how many became active users. To our knowledge, this is the first study to provide a precise estimate of how likely people are to use a rehabilitation app when presented with the opportunity in a clinical setting. This, in turn, allows for evaluation of app uptake and if changes in recruiting method or the app design affect use. We consider such data important in understanding the expanding and unregulated

**Table 5. Joint display of quantitative adherence data and qualitative explanatory findings.**

| QUAN outcome | Theme | Sub-theme | Qual outcome (participant) | Interpretation | Impact |
|---|---|---|---|---|---|
| **Uptake:** 79% (38 of 48) completed at least one session in week 1. | III | *Diagnostic and prognostic action* | *I definitely had an expectation that they would take x-rays which they also did and when they then determined that nothing was broken, I was given a bandage dressing and instructions about how to treat this sort of swelling, so that was strictly by the book.* (ID 7) | Patients expect a diagnostic focus when visiting the ED, and do not expect or demand a focus on rehabilitation. | Decreased uptake |
| | III | *Diagnostic and prognostic action* | *I thought that they [the Emergency Department] should be able to see on the x-ray if it would take a month or two months.* (ID 12) | Patients believes that time to full recovery can be predicted from the diagnostics but did not consider their involvement as part this prognostic. | Decreased uptake |
| | III | *Treatment and preventive action* | *I don't know if it was a health professional, it was of course, probably, a doctor, but there was also an intern on the side, but I still think the level of information were fairly low you could say. I had expected some more advice and guidance and a more extensive explanation on what the injury was.* (ID 12) | Patients felt insufficiently informed about their ankle sprain and did not feel guided in the following clinical course | Decreased uptake |
| | III | *Provider integrity* | *It seemed sort of verified. Like it wasn't some kind of scam-app, who would be like "try this" and then there would be all kind of commercials and premium stuff and whatever that you end up spending money on. This felt trustworthy and verified.* (ID 13) | Health personnel is seen as important influencers when patients are exposed to the app. | Increased uptake |
| | III | *The user as independent searcher* | *I don't use apps much so I would never have figured to go and search for an app that could help me.* (ID 5) | The ED may be an important setting to present an app solution, since patients may not independently search themselves. | Increased uptake |
| **Adherence:** Minimum 50% (24 of 48) completed at least 1 exercise per week in the first 6 weeks | I | *Usability* | *I thought it was good because I could do it at work if I had 5 minutes to spare. First of all, it didn't take very long, and you were able to do it everywhere. That was a major plus, that you were able to do it everywhere.* (ID 17) | Short duration made the program easy to commence | Increased adherence |
| | I | *Usability* | *I didn't fancy those exercises where you had to lay on a madras, because then you have lie down and then you have to find the madras and where should it lie? . . . I liked those exercises where you just use a chair in front of you, and then either you have to go stretch on it, or you have to hold your balance, or sit on it and do an exercise, I think that's simple.* (ID 2) | Even basic requirements may decrease the usability. | Decreased adherence |
| | II | *Process statistics* | *I think it's fine, then I can see that now I have done 10% now I have done 20%, which I really like. That is if I progress you know. I actually really like It . . . It's like running on a treadmill where you can see how far you've run. I really like it.* (ID 4) | Simple statistics gave a feeling of being part of a progress | Increased adherence |
| | I | *The app's exercise level and ability to adapt* | *I actually think it was fine to begin with. So, to begin with, it was actually fine, there was nothing that bothered me, it wasn't until I tried it, yeah, I think maybe I had been doing it for about a week and had been doing it those three, four sessions that I felt "okay this, doesn't challenge my ankle enough for it to help with my rehabilitation.* (ID 13) | Inappropriate starting level and/or progression gave frustrations when performing exercises | Decreased adherence |

*(Continued)*

**Table 5.** (Continued)

| QUAN outcome | Theme | Sub-theme | Qual outcome (participant) | Interpretation | Impact |
|---|---|---|---|---|---|
| | I | *The app's exercise level and ability to adapt* | *For over a month I've been in phase two and I simply don't understand it and I actually get pretty mad when it says "you have completed" and then it's still at 90%, I should have finished phase three by now but I still can't get to it."* (ID 4) | Patients felt stagnated when they did not understand progression in the app | Decreased adherence |
| | I | *The app's exercise level and ability to adapt* | *I don't know why it didn't progress . . . I answered that the exercise was hard, but that doesn't make it bad. It is like doing strength training then you also must push yourself to increase you level and that is hard but doesn't make it bad.* (ID 2) | If participants felt, they could not communicate around key concepts with the app they became frustrated. | Decreased adherence |
| | I | *The app's exercise level and ability to adapt* | *It seemed that you had to go through the first step and then the second and the third. You couldn't just skip to step three if you wanted something different or more challenging.* (ID 13) | Lack of autonomy in progression may negatively affect adherence | Decreased adherence |
| | I | *Ankle symptoms and expectations.* | *I thought it would take a month. Especially because I heal very well (laughing), maybe not as well as previously but I still thought that it probably would take about a month before I was completely healthy again.* (ID 12) | There was a general belief that an ankle sprain would recover spontaneously within few months. | Decreased adherence |
| | I | *Ankle symptoms and expectations.* | *The problem was that it started to go well with my ankle . . . I don't think that I was injured enough to keep the motivation. It's like people with back pain. As soon as the pain is gone, they don't do their exercises and I think it was the same that happened to me.* (ID 17) | When symptoms from the ankle sprain decreased, they felt less motivated to exercise | Decreased adherence |
| | I | *Influence of work and leisure time* | *It was really just returning to work and the chores of daily living. It didn't have anything to do with the exercises or the app.* (ID 7) | When participants became able to work, perform daily chores or travel they lost motivation for exercising. | Decreased adherence |
| | II | *Process statistics* | *I think it's fine, then I can see that now I have done 10% now I have done 20%, which I really like. That is if I progress you know. I actually really like It . . . It's like running on a treadmill where you can see how far you've run. I really like it.* (ID 4) | Simple statistics gave a feeling of being part of a progress | Increased adherence |
| | II | *Reminder function* | *Well I basically turned it off, because I decided not to receive the push-notifications which is what I generally do with all apps or programs I install, so I don't get bombarded with various pointless information. Here, it would actually have been fine for me.* (ID 7) | Reminders did not seem to be an important function in the app, but many had turned it off already from the beginning. | No influence on adherence |
| | II | *Exercise comprehension* | I was surprised how easy it was to follow the instructions in the exercise videos. I really think this is relevant because it was so easy to just start the program. (ID 5) | The videos were a strong contributor to exercise comprehension. | Increased adherence |
| | II | *Views on visual expression* | I actually think it was really good with a naked room without anything but the things that were being used, a chair, a table, a mattress. You know a room with nothing, not even on the wall. I think that was really good. (ID 2) | A plain video expression seemed to give integrity to the exercises. | Increased adherence |

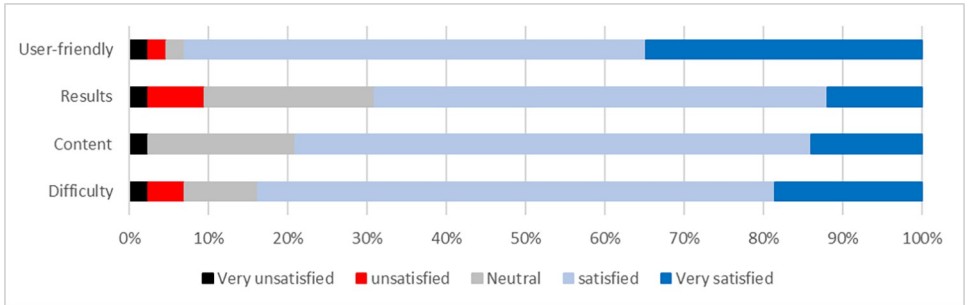

**Fig 5. Satisfaction scores for different app use items.**

field of health apps [14]. Since this is the first study to evaluate the delivery of an exercise app for ankle sprains in an ED setting, it is difficult to compare the uptake data to many other studies. Vriend et al. [29] did evaluate the uptake of an exercise app for people with previous ankle sprains when advertised in national media and on sports facilities. They estimated that the app was downloaded by less than 2.6% of their targeted population despite intensive marketing, and that only 62% of those who downloaded the app became active users. Though the authors were not able to determine the percentage of the targeted population that became aware of the app´s existence, they concluded that a "marketing" type of strategy may not be the optimal method of implementing an evidence-based app. Compared to the estimated 2.6% referenced above, the 10% active users in our study was better, however, it is difficult to consider 10% uptake a success. The higher uptake-level in our study indicates that the direct delivery of the app from a health care professional in the ED can encourage more people to download the exercise program. This was further supported by our qualitative data showing that the app, when given by a health care professional, seemed trustworthy and this had influenced participants to download the app; a finding that aligns with a previous study where people stated they were more likely to use an app if it was endorsed by a health care professional [30]. It was, however, beyond the scope of this study to investigate how the health care professionals delivered the app information and their beliefs towards it, and whether this affects uptake to a high degree. Interestingly, the interviews also revealed that several participants felt insufficiently informed about their injury when leaving the ED. This might indicate that more adequate information from health care professionals about the consequences of LAS and recommended exercise rehabilitation could prompt more patients to use the app and health care professionals in the ED could have a great opportunity to influence people's behavior by providing such information in a clinical setting.

The active users completed a median of 5.5 exercise sessions. Most participants became active in the first week, but 20% started after the first week. In general, adherence declined through the study period and after 2 weeks less than half of the users performed 2 or more weekly sessions. After 9 weeks around 20% continued to use the app through the 8 months study period. Because this study was exploratory, we did not pre-specify a threshold for acceptable adherence. This is important because studies have found that a home-based exercise

**Table 6. Weekly ankle stability scores.**

| Week | 1 | 2 | 3 | 4 | 5 | 6 | 7 | 8 | 9 | 10 | 11 | 12 | 13 | 14 | 15 | 16 | 17 |
|---|---|---|---|---|---|---|---|---|---|---|---|---|---|---|---|---|---|
| Ankle stability | 4.5 | 5.7 | 6.5 | 6.8 | 6.9 | 7.4 | 7.5 | 7.8 | 7.9 | 8.2 | 7.9 | 8.1 | 8.1 | 8.6 | 8.6 | 8.6 | 8.7 |

The mean scores for subjective ankle stability (0–10 points) for each week. 0 = Very unstable, 10 = Completely stable.

program with 24 exercise sessions could reduce the risk of recurrent sprains by 35% [12,31]. If we consider 24 exercise sessions to be an acceptable adherence threshold (not considering time per session or time intervals between completed sessions) only 15% of the participants in the present study were adherent.

Despite the low adherence, the participants reported high satisfaction with the app although they did not use it much. Almost all participants would recommend the app to others, which is consistent with data from Vriend et al. [29]. Their app was also given high appraisal by its users even though they only completed, on average, 3.3 exercise sessions out of the recommended 24 in the app. One would think that they stopped exercising because they no longer felt restricted by their ankle sprain. However, when asked about this in the present study (if they felt recovered when they stopped using the app) the majority responded "No". Participants stated in the interviews that they had lost motivation when the symptoms declined to a level where they were able to manage daily tasks. From the clinical recovery data, we can see that crutches are predominantly used in the first two weeks. So, even though 80% of the participants after two weeks still suffered symptoms, it is likely that they could have started to work and participate in leisure activities and thus feel it less necessary to continue using the app. This may have contributed to why they stopped exercising with the app. The high satisfaction with the app–despite limited use–could be related to social desirability bias, that is, patients reporting what they expect would please the investigators.

The interviews revealed that participants in general expected a complete spontaneous recovery from the ankle sprain regardless of their actions. That people with ankle sprain believe the injury to be innocuous is anecdotally supported by several studies [1,4,10,32–34] but to our knowledge, this is the first study that has interviewed people on this perception. The perception that an ankle sprain is an innocuous injury may also be reflected in the reason for seeking medical attention at the ED, as participants primarily went to the ED because they were worried that the ankle had a fracture, not because they were worried about the consequences of a sprained–not fractured–ankle. How the perception of LAS as being an innocuous injury influences adherence to an app and an exercise program is largely unknown. But from the joint display of the qualitative and quantitative data, it seems that the focus of their ER visit was diagnostic and that when symptoms decreased, they lost motivation for exercise. This likely resulted in low overall exercise adherence but with paradoxically high satisfaction across the different app use items. The interviews revealed that several participants experienced the starting level to be either too difficult or too easy. They found it discouraging if the app did not match their expectations regarding exercise difficulty within a few completed sessions. The app was designed with a fixed starting level and a hierarchical development of exercises. The advantage of this design is that the progression can be matched to exercise guidelines and minimize the risk that users are presented with exercises that may cause them harm. The disadvantage is that some participants may need to complete several sessions before they reach a desired exercise difficulty, which may negatively impact adherence.

The app did provide the participants with an exercise solution they found easy to access and the short programs were perceived as easy to include in the daily routines. Both time restrictions and access to exercise opportunities have been found to be major barriers for physical activity among patients with musculoskeletal disorders [35]. It is interesting that despite the app-based exercise program resolved these two major barriers and it was highly appraised, it was not enough to substantially motivate our participants to exercise. Whether the low adherence was primarily due to a general opinion that ankle sprains are an innocuous condition, and/or the app-based exercise program–especially the starting level–is currently unknown. Further research is needed to evaluate how different recruitment methods, program designs and conditions affect adherence for app-based exercise interventions.

## Future app optimizing

The app-based exercise program enables users to perform exercises wherever and whenever. However, data from the interviews point out that usability of an exercise app may be more than just being ever-present in your pocket. Time to complete sessions, exercise materials, the need for changing into gym clothes or just getting down on the floor are all factors that may influence and limit the usability. Some demands can be necessary for the exercise program to ensure correct exercise form; however, it may enhance adherence if users could customize their program to fit not only their injury but also their exercise behavior. Giving users more control of their exercise program may also be a solution for users who feel that the starting level is far from what they want it to be. The exercise app seemed to have one or two sessions to match people´s expectations before they would quit the program. Since the app can make real-time changes dependent on user feedback, an improvement could be to include initial questions on people's functional disabilities (e.g. ability to stand or walk), so that it may guide the app to a more motivational starting difficulty. Furthermore, giving users the ability to skip difficultly levels would make the app more adaptable to both user expectations and day to day changes in symptoms.

It seems that the explanatory exercise videos are important, and they seem to make people feel secure in their exercise execution. The visual expression in the videos seems to influence the integrity of the app. Most of our participants seemed to prefer what they called the "clinical expression" used in this app with a bare room and a regular looking person, compared to more fitness-focused app with highly trained athletes and fast paced music. They wanted a person they could relate to. It is likely that different age groups, gender etc. prefer different video expressions and a personalized video could enhance motivation to perform exercises with the app as a partner.

With regards to the statistics in the app, we were surprised that participants did not consider them important for their motivation. It was surprising because many thought the process statistics presented after each completed session were a measure of their clinical recovery despite the fact it only reflected the program process. This is, however, similar to the finding from Liao et al. [36] who found that visual demonstration was perceived as the only important motivational factor among 52 app design features including reminder and statistical functions. A reason that the statistics are not found to be motivating could be that they are generic and not user tailored. A patient-specific goal setting may enhance users work towards those goals [37,38] and a scale like the Patient Specific Functional Scale [39] would be easy to include as an app feature. Finally–and also related to goal setting—the app could potentially benefit from some educational information initially about what an ankle sprain is, and how much rehabilitation exercise is needed for recovery. This should also include the importance of continuing exercise when symptoms decrease so that the risk of recurrent sprains is decreased. A standard user experience questionnaire could then be used for evaluation of any app changes made.

## Strengths and limitations

The adherence data in this study do not rely on feedback from participants and are therefore not affected by potential recall or reporting bias. This is a major study strength. Only exercises that are registered in the app are recorded, however. One participant described that after learning several of the exercises, she had performed them without opening the app.

The app collects feedback on individual exercises, which is much more detailed than just session collecting completion, which is a commonly used proxy measures for adherence [40]. From the data, it would be possible to identify possible exercises that participants experience too difficult or pain provoking at a certain stage. A limitation, however, it that the app does not monitor patients during the exercises and performance quality is not assessed.

In this study it was possible to elaborate the quantitative app-use data with qualitative data from the interviews and suggest possible explanations. This would not have been possible if only one methodological approach has been chosen. The mixed method approach in this study is described with regards to "Good Reporting for a mixed method study" (GRAMMS) [24] to increase transparency and the author group comprised of both quantitative and qualitative experts to obtain quality of each approach which is advocated for mixed method design [24].

## Conclusion

In this study, only few of the patients seen for an ankle sprain in the ED became active app users after they received information about a free app-based rehabilitation exercise program. Those who did, liked the app very much, but few completed enough exercise sessions to realistically impact clinical recovery. The ankle sprain was generally considered an innocuous injury that would spontaneously recover even though more than half the participants did not feel fully recovered when they stopped exercising, and a third experienced a recurrent sprain. To improve the care of these patients in the ED, we suggest that health-care personnel who asses acute ankle sprains should be aware of their importance in informing patients about the risk of prolonged symptoms and recurrent sprains, so that patients may have more realistic expectations on the clinical course. Recommendations from a health care professional in the ED to use an ankle sprain exercise program seems to carry more weight than similar recommendations given elsewhere, making the ED setting interesting from an implementation point of view.

## Supporting information

**S1 Table. Good Reporting of A Mixed Methods Study (GRAMMS) checklist.**
(DOCX)

**S2 Table. Exercise program.**
(DOCX)

**S1 Fig. Number of completed exercise sessions per participant.**
(DOCX)

**S2 Fig. Adherence by different grouping variables.**
(DOCX)

**S1 Text. Description of the exercise program.**
(DOCX)

**S2 Text. ICMJE disclosure forms for all authors.**
(PDF)

**S3 Text. Study protocol.**
(PDF)

## Author Contributions

**Conceptualization:** Kristian Thorborg, Jeanette Wassar Kirk, Thomas Bandholm.

**Data curation:** Jonas Bak, Mikkel Bek Clausen.

**Formal analysis:** Jonas Bak, Mikkel Bek Clausen, Jeanette Wassar Kirk, Thomas Bandholm.

**Funding acquisition:** Kristian Thorborg, Thomas Bandholm.

**Investigation:** Jonas Bak, Kristian Thorborg, Mikkel Bek Clausen, Finn Elkjær Johannsen, Jeanette Wassar Kirk, Thomas Bandholm.

**Methodology:** Jonas Bak, Kristian Thorborg, Mikkel Bek Clausen, Finn Elkjær Johannsen, Jeanette Wassar Kirk, Thomas Bandholm.

**Project administration:** Jonas Bak.

**Resources:** Kristian Thorborg, Mikkel Bek Clausen, Finn Elkjær Johannsen, Thomas Bandholm.

**Software:** Mikkel Bek Clausen.

**Supervision:** Kristian Thorborg, Jeanette Wassar Kirk, Thomas Bandholm.

**Validation:** Jonas Bak, Jeanette Wassar Kirk.

**Visualization:** Jonas Bak, Kristian Thorborg, Jeanette Wassar Kirk, Thomas Bandholm.

**Writing – original draft:** Jonas Bak, Kristian Thorborg, Mikkel Bek Clausen, Jeanette Wassar Kirk, Thomas Bandholm.

**Writing – review & editing:** Jonas Bak, Kristian Thorborg, Mikkel Bek Clausen, Finn Elkjær Johannsen, Jeanette Wassar Kirk, Thomas Bandholm.

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
