## [Decision Letter · Decision Letter 0]

26 Jan 2023

PDIG-D-22-00320

Using the app “Injurymap©” to provide exercise rehabilitation for people with acute lateral ankle sprains seen at the Hospital Emergency Department – a mixed method pilot study

PLOS Digital Health

Dear Dr. Bandholm,

Thank you for submitting your manuscript to PLOS Digital Health. After careful consideration, we feel that it has merit but does not fully meet PLOS Digital Health's publication criteria as it currently stands. Therefore, we invite you to submit a revised version of the manuscript that addresses the points raised during the review process.

Please submit your revised manuscript within 30 days Feb 25 2023 11:59PM. If you will need more time than this to complete your revisions, please reply to this message or contact the journal office at digitalhealth@plos.org. Please include the following items when submitting your revised manuscript:

We look forward to receiving your revised manuscript.

Kind regards,

Jasmit Shah

Guest Editor

PLOS Digital Health

Journal Requirements:

1. Please send a completed 'Competing Interests' statement, including any COIs declared by your co-authors. If you have no competing interests to declare, please state "The authors have declared that no competing interests exist". Otherwise please declare all competing interests beginning with the statement "I have read the journal's policy and the authors of this manuscript have the following competing interests:"

a. Please clarify all sources of funding (financial or material support) for your study. List the grants (with grant number) or organizations (with url) that supported your study, including funding received from your institution. 

b. State the initials, alongside each funding source, of each author to receive each grant.

c. State what role the funders took in the study. If the funders had no role in your study, please state: “The funders had no role in study design, data collection and analysis, decision to publish, or preparation of the manuscript.”

d. If any authors received a salary from any of your funders, please state which authors and which funders.

3. We ask that a manuscript source file is provided at Revision. Please upload your manuscript file as a .doc, .docx, .rtf or .tex.

4. In the online submission form, you indicated that :National legislation prohibit public sharing of a the full data set. Reviewers and others may obtain access to the data by request, and after a data processing agreement has been signed.". All PLOS journals now require all data underlying the findings described in their manuscript to be freely available to other researchers, either 1. In a public repository, 2. Within the manuscript itself, or 3. Uploaded as supplementary information.

Additional Editor Comments (if provided):

Reviewers' comments:

Reviewer's Responses to Questions

**Comments to the Author**

1. Does this manuscript meet PLOS Digital Health’s publication criteria? Is the manuscript technically sound, and do the data support the conclusions? The manuscript must describe methodologically and ethically rigorous research with conclusions that are appropriately drawn based on the data presented.

Reviewer #1: Yes

Reviewer #2: Yes

Reviewer #3: Yes

2. Has the statistical analysis been performed appropriately and rigorously?

Reviewer #1: Yes

Reviewer #2: Yes

Reviewer #3: N/A

3. Have the authors made all data underlying the findings in their manuscript fully available (please refer to the Data Availability Statement at the start of the manuscript PDF file)?

Reviewer #1: Yes

Reviewer #2: No

Reviewer #3: Yes

4. Is the manuscript presented in an intelligible fashion and written in standard English?

Reviewer #1: Yes

Reviewer #2: Yes

Reviewer #3: Yes

5. Review Comments to the Author

Reviewer #1: In this manuscript, the authors presented excellent research that provides readers with clinically useful information that will be useful in the management of acute lateral ankle sprains. This work demonstrates the effectiveness of specific apps to provide exercise rehabilitation for people with this injury. Moreover, the research investigates important points of app use (uptake, retention, exercise adherence and satisfaction)

Reviewer #2: This paper evaluates the feasibility and primary clinical effectiveness of the Injurymap© exercise app among patients with lateral ankle sprains seen in the Emergency Department at Copenhagen University Hospital. Sixty participants were enrolled into the study, the authors found low adherence to the app, however participants were highly satisfied with the app and would recommend it to others. The authors also integrated qualitative results from semi-structured interview with a subset of the recruited participants to contextualize the engagement results. Through these interviews they identified that most participants considered ankle sprains as an innocuous injury that would recover by itself and did not feel well informed about recovery from Emergency Department healthcare professionals. The authors also identified enablers and barriers to app engagement through qualitative component. This paper was clear and succinct, written very well and reports important findings which would be relevant to the PLOS Digital Health’s readership. Please find below my comments and some minor revision recommendations:

Introduction 

- Very clearly laid out, I really appreciated the structure, how concepts were explained, how the gaps in current research were identified and the aims were presented. 

- Found figure 1 especially useful to understand the explanatory sequential mixed method cohort study design.

- First recommendation, it would be helpful to move “Study Design” section into the methods instead of it being in the “Introduction” section, it can be presented before “Study setting”

Methods

- Cleary laid out and detailed. 

- Well thought out analysis plan and explanation on why original analysis plan wasn’t completed (i.e. the shift to descriptive analyses). 

- First recommendation, in both the methods and results authors refer to participants as “people” – if possible, I would suggest call individuals who participated in the study as “participants” throughout the manuscript. 

- Second recommendation, on page 8 in the “material and outcome assessment” section would clarify “email” in the following sentence: 

o Participants received emails containing a link to their personal online questionnaire.

- Third recommendation, for both the qualitative and quantitative analysis it would be helpful to report what software was used to perform the analysis. 

Results

- Cleary laid out and sufficient detail is provided. 

- Found table 5 integrating the qualitative and quantitative results very informative and got the point across well. 

- First recommendation, for table Table 1 and 2, it would be helpful if a more detailed title was provided that include the sample size (e.g., Baseline characteristics of participants with LSA [n=60])

- Second recommendation, on page 25 for figure 5, I believe there was an error and the sentence was to say “did not” understand:

o Patients felt stagnated when they did not understand progression in the app

Discussion 

- Great discussion, overall, the authors were very transparent about their findings and did a really great job at contextualizing the results and providing thoughtful discussion about reasons for poor adherence and engagement

- The future app optimization section was well thought out and realistic in terms of implementation.

Reviewer #3: This submission evaluates the use of an app to support patients with physical exercises after a LAS injury. The app is supposed to use "machine-based learning" which I take to mean machine learning, possibly to provide real-time feedback.(reading on, I am still not quite clear what this refers to.) However, that is not quite clear. The app in question is InjuryMap which is not described in enough detail so that the study results can be put in context.

The study design is a mixed methods study without a control comparison so it is really a usability evaluation of the app in addition to clinical outcomes. Figure 1 is confusing; it's not clear how the qual outcomes resulted from the quan: app-use and how the baseline questionnaire was applied before the intervention (which is use of the app?). It would be great to mark up Figure 1 with the number of participants that took part in the various branches. How did the distribution of the recruitment packs figure into the process?

There are several standard and validated user experience questionnaires available, why did the study not use any of them? However, the study methods are generally very good.

In terms of results, theer are some sobering figures hidden in the supplementary material that probably should be highlighted in the main submission: adherence is really low and there are 2-3 patients who do a lot of exercises, and then there is a very long tail of people not doing much. The median is 5.5 exercises over the course of 4 months! In terms of physical rehabilitation this is not enough. I think the satisfaction also needs to be taken with a pinch of salt: I think they were being nice about the feedback when they clearly didn't use the free app. It is also not a surprise that there is no reliable clinical outcome as they didn't adhere to the exercises.

The discussion of the results is interesting but possibly raises even more questions than are answered. Recruitment and uptake is obviously of prime importance but it is well-known that healthcare professionals play a key role in making better choices. I was really hoping that the qualitative results would point the way to improving the app, or other apps that deliver treatment. I found it also frustrating that the discussion of the user experience didn't link back to the results.

spelling issues:

- p. 3: an exercise session

- p. 6: is comprised of

-p. 18: questionnaire

6. PLOS authors have the option to publish the peer review history of their article (what does this mean?). If published, this will include your full peer review and any attached files.

**Do you want your identity to be public for this peer review?** For information about this choice, including consent withdrawal, please see our Privacy Policy.

Reviewer #1: No

Reviewer #2: No

Reviewer #3: No

---

## [Editor Report · Decision Letter 1]

27 Feb 2023

Using the app “Injurymap” to provide exercise rehabilitation for people with acute lateral ankle sprains seen at the Hospital Emergency Department – a mixed-method pilot study

PDIG-D-22-00320R1

Dear Dr. Bandholm,

We are pleased to inform you that your manuscript 'Using the app “Injurymap” to provide exercise rehabilitation for people with acute lateral ankle sprains seen at the Hospital Emergency Department – a mixed-method pilot study' has been provisionally accepted for publication in PLOS Digital Health.

Best regards,

Jasmit Shah

Guest Editor

PLOS Digital Health